# A Global Remote-Sensing Assessment of the Intersite Variability in the Greening of Coastal Dunes

Petya G. Petrova, Steven M. de Jong and Gerben Ruessink *

Department of Physical Geography, Faculty of Geosciences, Utrecht University,
P.O. Box 80.115, 3508 TC Utrecht, The Netherlands; petrova.petya@gmail.com (P.G.P.); s.m.dejong@uu.nl (S.M.d.J.)
* Correspondence: b.g.ruessink@uu.nl

**Abstract:** In recent decades, the vegetation on many coastal dunes has expanded spatially, which is attributed, among other things, to global-scale climate change. The intersite variability in this dune greening has not yet been substantially investigated, nor is it known whether it is consistent with intersite variability in climate change. Therefore, the objectives of this work were firstly to quantify and analyse the change in vegetation cover from multitemporal NDVI time series at a large number (186) of dune fields worldwide, calculated from Landsat satellite imagery available between 1984 and 2021 and secondly, to correlate the identified trends with trends in the main climate variables influencing vegetation growth (temperature, precipitation and wind speed). We show that greening is strongest in cool temperate climates (35° to 66.5° north/south latitudes) and that the rate of greening is accelerating at many sites. We find no dependence between the rate of greening and the local temporal change in temperature, precipitation and/or wind speed. Based on existing literature, sand supply and anthropogenic activities are discussed as possible reasons for the absence of a clear global relationship between variability in dune greening and climate change.

**Keywords:** coastal dunes; greening; NDVI; time series decomposition; climate change; Landsat imagery; geomorphological stabilization





## 1. Introduction

Dunes are globally present aeolian features associated with sandy, wave-dominated coastal landscapes with an ample sediment supply and sufficiently strong onshore winds [1–3]. They arise from the interaction between wind, sand and vegetation cover and have an important role in the sediment budget of the beach-dune system [2,4]. Providing coastal stability and protecting the hinterland from marine flooding is widely regarded as the most important ecosystem service of coastal dunes to humankind. This is particularly important nowadays in view of climate change with its associated rise of the mean sea level and the increasing threat from more frequent and more destructive storms. The ecosystem services of coastal dunes also include natural values. Dunes contain highly diverse habitats with rare plants and animal species, therefore holding a high biodiversity value [3,5]. Dunes also contain natural resources, such as sand and drinking water [6,7] and have a high recreational potential [8].

Many local to regional studies have illustrated that coastal dunes have stabilized over the last decades due to the spatial expansion of vegetation [9–14], called greening. Coastal dune mobility is driven primarily by three groups of factors: (1) climate (wind regime, precipitation, temperature); (2) sediment supply and availability and (3) extent of vegetation cover [1,15,16]. Vegetation growth and spatial expansion can be climate-driven (e.g., reduced wind speeds, increased temperature) and anthropogenic (e.g., planting activities, rabbit diseases, nitrogen deposition from air pollution) [11,16]. Other anthropogenic disturbances can, in contrast, reduce vegetation productivity and hence stimulate dune mobility (e.g., mowing, grazing, vegetation removal [17–20]). There is much discussion

about the consequences of greening for the ecosystem services of coastal dunes. On the one hand, vegetation may make the dunes more resistant to erosion by storm waves and hence improve their protective capacity [21–23]. On the other hand, the more pronounced vegetation cover minimizes dune mobility and blocks landward aeolian sand transport [24,25]. This may prevent dune fields from migrating inland and growing vertically under sea-level rise, threatening their existence under global climate change [26]. Furthermore, the lower mobility can reduce biodiversity [27] and hence cause the dunes to be less able to withstand stressors imposed by climate change [28].

The greening of coastal dunes on a global scale has recently been discussed by Jackson et al. [22] and Gao et al. [29], among others, with respect to the major driving forces behind the stabilization process. Based on a remote-sensing study of 17 study sites with minimal human activities, ref. [22] argued that greening was consistent with global-scale changes in the main climatic variables (i.e., increasing temperature and precipitation, combined with decreasing windiness) and atmospheric composition or deposition of various pollutants and nutrients ($NO_2$, $CO_2$, N, P, etc.). They did not explore whether intersite variability in the strength of the greening trend was related to a local or regional change in climate conditions. Based on a literature review, ref. [29] classified dune mobility trends at 176 sites worldwide into stabilization (or greening), mobilization and little change, and confirmed [22]'s finding that coastal dunes were greening at a global scale. They further argued that the greening was primarily due to human interventions, including land-use changes (urbanization, tourist development), reduced sediment delivery to dunes by river dams or other constructions, grass planting and afforestation. Ref. [29] did not quantify intersite variability in coastal dune greening.

The objectives of this paper were first to quantify intersite variability in dune greening and second, to elucidate whether this variability was consistent with intersite temporal changes in the main climate factors affecting dune mobility. To reach the first objective, we performed a global (186 sites) remote-sensing assessment of dune dynamics based on time series of the normalized difference vegetation index (NDVI) quantified from Landsat 4/5/7/8 satellite images between 1984 up to and including 2021. For the second aim, we coupled the NDVI time series with ERA5 reanalysis time series of the main climatic variables relevant to dune greening (temperature, wind speed, rainfall). The remainder of this paper, which is based on Petrova [30], is organized in four sections. Section 2 describes the selected dune sites and illustrates how the NDVI time series were computed and analyzed. This section also introduces the time series of the climatic variables and their analysis. The results of our work are presented and discussed in Sections 3 and 4, respectively. The main conclusions are stated in Section 5.

## 2. Materials and Methods

### 2.1. Selected Coastal Dune Sites

The selection of coastal dune sites was determined to a great extent by the recent reviews and discussions on dune mobility by [22,29]. We added several sites to widen the selection in terms of geographical locations, climate regions and human interventions, following visual inspection of different coastal dune systems worldwide in Google Earth Pro. The first selection criterion was the natural aspect of the dunes. We included dune sites with a natural origin with no to moderate direct human interventions. In our work the term "moderate" relates to sites that were restored (vegetation planting or removal) and sites where tourists are allowed to walk on the surface. We excluded natural dunes that experienced land use changes by agricultural or urban expansion as well as artificial (i.e., engineered) dunes, e.g., [31]. The second criterion was the inland extension of the dunes; it should be sufficiently large to avoid limitations of the relatively low resolution of Landsat images (30 × 30 m).

Based on these criteria, a total of 186 dune fields were selected for the current study. Figure 1 illustrates their geographical location superimposed on a map of the world climate regions. Supplementary Table S1 shows the site names, together with the latitudes and

longitudes. The selected locations cover all ecozones in the world except Antarctica and are distributed geographically as follows: 69 in Europe (e.g., [9,11,18,20,32–41]); 33 in South America (e.g., [42–46]); 31 in North America (e.g., [47–54]); 20 in Africa (e.g., [55–57]); 29 in Oceania (e.g., [12,19,58–61]); and 4 in Asia (e.g., [46,62]). In contrast to [22], coastal dunes from arid to semi-arid areas were also selected here. Moreover, for the sites with dry tropical settings already reviewed by [29], we also included, for example, the chevrons on the Island of Madagascar [57] as well as the Guerrero Negro backbarrier dunefield in Baja California, Mexico [63]. Examples of selected sites with human interventions can be found along the Northwest European and New Zealand coasts, where locally, vegetation has been removed to stimulate aeolian dynamics and dune mobility [19,20].

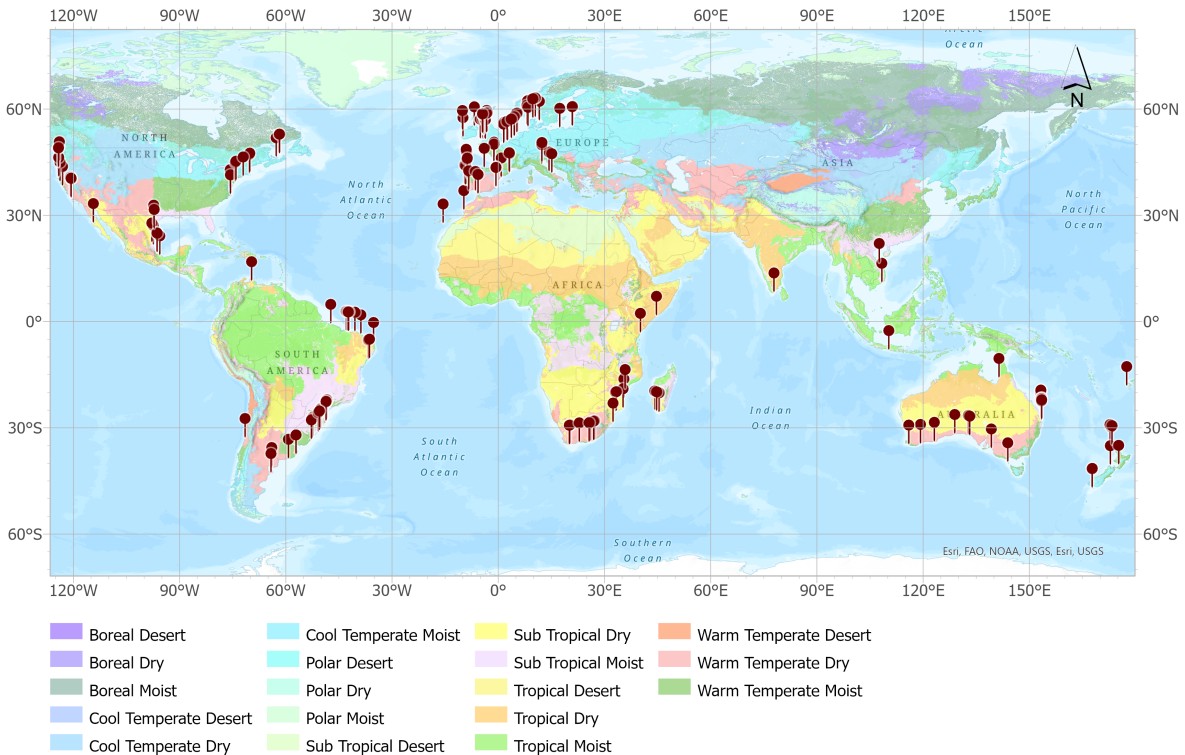

**Figure 1.** Location of the selected dune sites superimposed on a map of the world climate regions (18 classes [64]). The climate regions are produced as a geospatial integration of the world temperature and moisture domains. Source: Esri, USGS, TNC (World Terrestrial Ecosystems Pro Package—Overview). The map was prepared in ArcGIS Pro 2.6.0.

For each site, a region of interest (ROI) was defined; Figure 2 shows two examples. The ROI included the entire dunefield when it was relatively small and had a clear perimeter (e.g., Figure 2a). Otherwise the ROI was a subset of the dunefield (e.g., Figure 2b) but always included the dune closest to the ocean as well as further inland located dunes. The polygon defining the outline of the ROI was drawn by visual interpretation based on high-resolution Google Earth Pro images. At most sites the beach–dune boundary was the most seaward location of the vegetation, see for an example Figure 2a. This boundary was more approximate at sites with sparse vegetation (e.g., Figure 2b). The median ROI area amounted to 3.5 km$^2$, with the 10th and 90th percentiles being 0.42 and ≈35 km$^2$, respectively. The smallest and largest areas were 0.03 km$^2$ (Marina di Ravenna, Italy [35]; Figure 2a and ≈275 km$^2$ (the transgressive Lençóis Maranhenses dunefield in NE Brazil [45]; Figure 2b), respectively. The area of each ROI can be found in Supplementary Table S1.

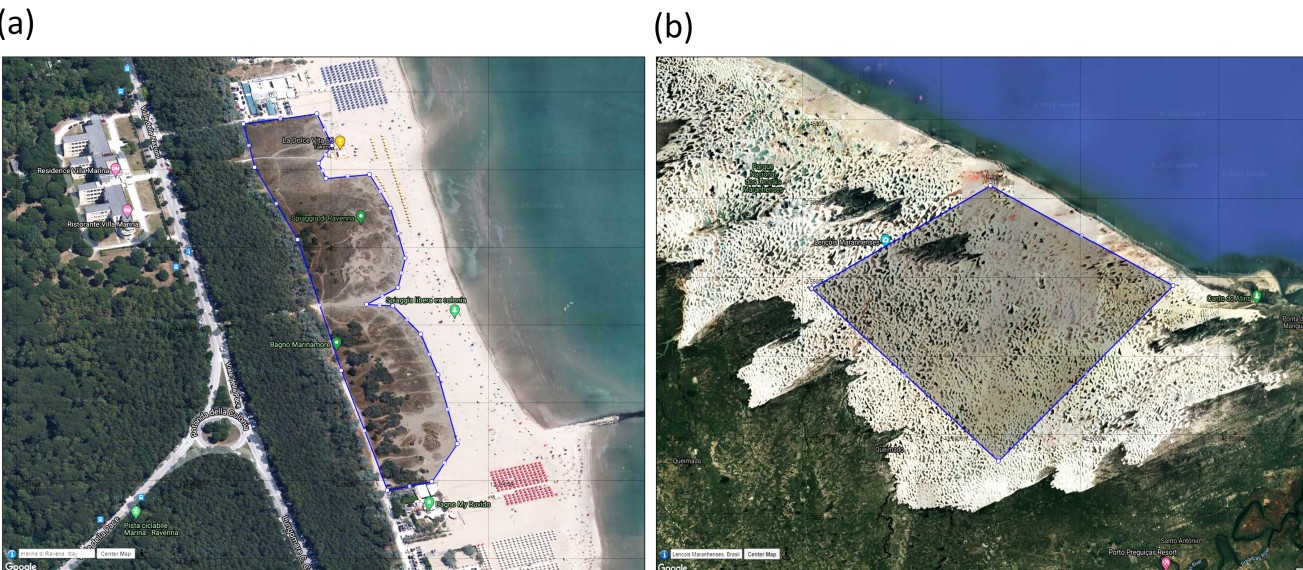

**Figure 2.** Examples of region of interest (ROI; shaded area bounded by blue line with white dots) for selected coastal dunefields: (**a**) Marina di Ravenna beach, Italy; and (**b**) Lençóis Maranhenses in NE Brazil. Source: Google Earth Pro 7.3.4.8248.

### 2.2. NDVI Time Series

To assess temporal changes in greenness, a time series of the normalized difference vegetation index, NDVI, was computed and downloaded for each site using the Climate Engine research App (https://app.climateengine.com/climateEngine, accessed on 24 February 2023 [65]). The Climate Engine app uses Google's parallel cloud-computing platform, Google Earth Engine [66], for on-demand processing of satellite imagery in real time. Each NDVI value is the average over the site's ROI, where the underlying NDVI maps are computed from the Landsat 4/5/7/8 surface reflectance data (Collection 2, Tier 1) in the red (Red) and near-infrared (NIR) bands ($NDVI = (NIR - Red)/(NIR + Red)$), with $30 \times 30$ m spatial resolution. The time period of each series is from 1984 up to and including 2021. Healthy vegetation is characterized by a low reflectance in the visible range of the spectrum and a high reflectance in the NIR range, and hence a dense green vegetation shows high NDVI values (0.6–0.9). Shrubs and grassland have intermediate positive values (0.3–0.5), and bare soil (e.g., dune sand) or very sparse vegetation have low positive values (0–0.3). Water bodies and clouds typically result in negative NDVI values. The greening of a dune site corresponds to more vegetated pixels in the ROI over time and is thus reflected by a statistically significant positive NDVI trend.

Prior to performing any statistical time series analysis, all NDVI signals were visually inspected and initial years with scarce data were, if present, removed. All negative NDVI values, probably due to cloudy images that were not classified as such and hence retained in the NDVI computation, were also removed. Local outliers were then identified and removed, with an NDVI value identified as outlier when it was more than three scaled median absolute deviations (MAD) away from the median. The scaled deviation was formulated as $A \; med(|NDVI - med(NDVI)|)$, where *med* is the median value and the scaling factor *A* equals approximately 1.4826. The outlier detection was performed with a moving-window procedure using a window length of 5 data points.

The cleaned NDVI time series of the present study contained trends and seasonal variations. An example of a cleaned NDVI time series to visualize this is given in Figure 3a. The interannual trends reflect greening (positive trend) or "browning" (negative trend; here, more sand pixels over time in the ROI), while the seasonal (intra-annual) variations are predominantly due to cycles in vegetation phenology ("growing season"). The series in

Figure 3a illustrates that the trends are not always monotonically increasing or decreasing but may vary in time in terms of direction and magnitude; in the example, the trend is positive over the first decade and negative over the final decade, with a relatively stable period in between. While the trend change in Figure 3a appears to be gradual, a limited number of other series showed a trend jump (breakpoint) as a result of a sudden change in the dune vegetation cover, possibly due to a severe storm. The amplitude of the seasonal variations can also change over time, as is clear from Figure 3a with the lowest amplitudes after 2014.

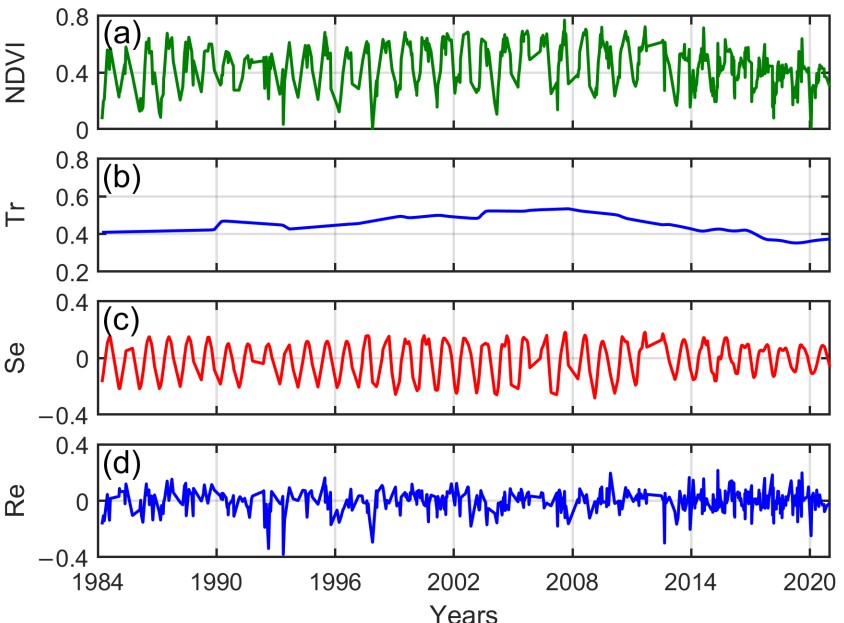

**Figure 3.** (**a**) NDVI time series at South Kennemerland, the Netherlands and its JUST decomposition in the (**b**) trend Tr, (**c**) seasonal Se and (**d**) remainder Re components. The decomposition shown in (**b**–**d**) was based on the chosen automated settings (see text), that is, a window size $M = 90$; a window translation step $\delta = 34$; thresholds for the magnitude *mag_th* and direction *dir_th* of the jumps of 0.2; and the minimum time difference between jumps *jump_th* = 2 years. No jumps were detected in this NDVI time series.

To study NDVI trends, including potential nonstationary behaviour, it is important that they are accurately extracted from the entire NDVI series (e.g., [67]). We adopted the open-source Jumps Upon Spectrum and Trend (JUST) software tool [68,69] to decompose the NDVI series in their trend Tr, seasonal Se and remainder Re components,

$$\text{NDVI}(t) = \text{Tr}(t) + \text{Se}(t) + \text{Re}(t), \tag{1}$$

where $t$ is time. Note that the temporal mean is contained in Tr. JUST was preferred over other decomposition methods (e.g., [70,71]) because it can handle unequally sampled nonstationary time series (as is the case here) without the need to resample them to a regular time grid. The decomposition algorithm in JUST is the antileakage least-squares spectral analysis [72], which operates with a moving-window approach. The window has a fixed size of $M$ observations and is moved through the series with a step of $\delta$ observations, where both $M$ and $\delta$ need to be set by the user. As detailed in [69], other user-defined parameters include thresholds for the absolute value of the jump magnitude (*mag_th*) and direction (*dir_th*), and the minimum time difference between jumps (*jump_th*). Based on recommendations in [69] and a series of tests reported in [30], $M$ was set to the total number of observations in the final three years and $\delta$ to the largest number of annual observations in these three years. The jump thresholds were set to rather high values (*mag_th* = 0.2; *dir_th* = 0.2; *jump_th* = 2 years) compared to their defaults to reduce the

chance of detecting an unrealistically large number of small jumps, while still permitting us to detect changes in the trend direction over the period of interest (1984–2021). The values were determined empirically [30] but followed [69]'s recommendation for JUST's automation capabilities. The decomposition of the example NDVI in Figure 3a with these settings is given in Figure 3b–d. These series confirm our earlier visual interpretation of Figure 3a, including the change in trend (Figure 3b) and seasonal amplitude (Figure 3c) over time.

The NDVI trend series computed with JUST were subsequently analysed with the nonparametric Mann–Kendall test (e.g., [73]) to determine the strength and direction of the trend and with the nonparametric Sen's method [74] to quantify its slope, $Q_{NDVI}$. A slope is statistically significant if the two-sided probability of the trend (*p*-value) does not exceed the chosen level of significance $\alpha$, beyond which an observed change can be considered a random occurrence. In the present work, the trend was evaluated against $\alpha = 0.01$. We also imposed that the magnitude of the NDVI change between 1984 and 2021 should exceed 1%, otherwise vegetation was labelled as "stable". A "stable" result thus implied either $p > 0.01$ or that the magnitude of the detected NDVI change was less than 1% over the study period. Finally, we note that we use the phrase "mobilizing" rather than "browning" to express that a negative NDVI trend implies a significant increase in the number of sand pixels (i.e., in aeolian dynamics) and not in a deterioration of the vegetation greenness.

### 2.3. Climate Data

The climate data for all 186 sites were also obtained from the Climate Engine research app. They consisted of ERA5 reanalysis data for the temperature (daily averages at a 2 m height; $T$ (°C)), total precipitation (daily sums; $P$ (mm/day)) and wind speed (daily averages at a 10 m height; $W$ (m/s)) for the period 1984–2021. ERA5 is a global climate reanalysis data set by the European Centre for Medium-Range Weather Forecasts, with a spatial resolution of 24 km ($0.25° \times 0.25°$) and a time coverage since 1979. The Mann–Kendall test was applied to all daily $T$, $P$ and $W$ series to identify statistically significant trends over time ($\alpha = 0.01$); Sen's method was subsequently applied to quantify slopes, denoted $Q_T$, $Q_P$ and $Q_W$, respectively.

## 3. Results
### 3.1. NDVI *Variability*

The spatial distribution of the NDVI trends based on the Mann–Kendall test is provided in Figure 4. Of the 186 sites, 162 (87.1%) became greener, 17 (9.1%) mobilized, and 7 (3.8%) remained stable over the study period. Figure 4 thus reconfirmed a clear dominance of the greening trends globally. The magnitude of change, $Q_{NDVI}$, is shown in Figure 5, in which the positive $Q_{NDVI}$ were grouped to roughly depict (1) slowly increasing ($0 < Q_{NDVI} \leq 0.003$ per year); (2) moderately increasing ($0.003 < Q_{NDVI} \leq 0.006$ per year); (3) rapidly increasing ($0.006 < Q_{NDVI} \leq 0.009$ per year); and (4) very strongly increasing ($Q_{NDVI} > 0.009$ per year) greenness. The strongest positive trends were identified at several locations: Ovari dune site, Southeast India, Thua Thien, Vietnam and Parangritis, Indonesia, as well as on the eastern Australian coast and in New Zealand. Rapid increases in greenness were also found in NW Europe. Figure 6 aggregates the geographical distribution in Figure 5 by showing the frequency of the four $Q_{NDVI}$ as a function of latitude, with latitude reflecting the regional variability of the climatic drivers, such as rainfall, temperature and wind regime (e.g., [39]). A large percentage of sites in the temperate zone, defined here between 35° and 66.5° north/south latitudes, fell in the range of moderate slopes (45 out of 85 sites; 53%) and less in the lowest slope range (35%). Dunes from the subtropics (between 23.5° and 35° north/south latitudes), on the other hand, mostly experienced a slowly increasing greenness (25 out of 52 sites; 48%) and to a lesser extent moderate increases (34%). The six sites with the highest $Q_{NDVI}$ were from both tropical regions (India, Indonesia and Vietnam) and subtropical regions (Brazil and Australia). Finally, the 17 cases with $Q_{NDVI} \leq 0$ per year included some of the few dune systems in Europe that

are mobile on a large scale today, such as the Dune du Pilat in France (the tallest sand dune in Europe), Valdevaqueros and Bolonia dunes in Spain, and the cliff-top Rubjerg Knude dune in Denmark, but were also found in Africa (e.g., Ampalaza Chevron in Madagascar) and Asia (e.g., Phan Thiet, Vietnam).

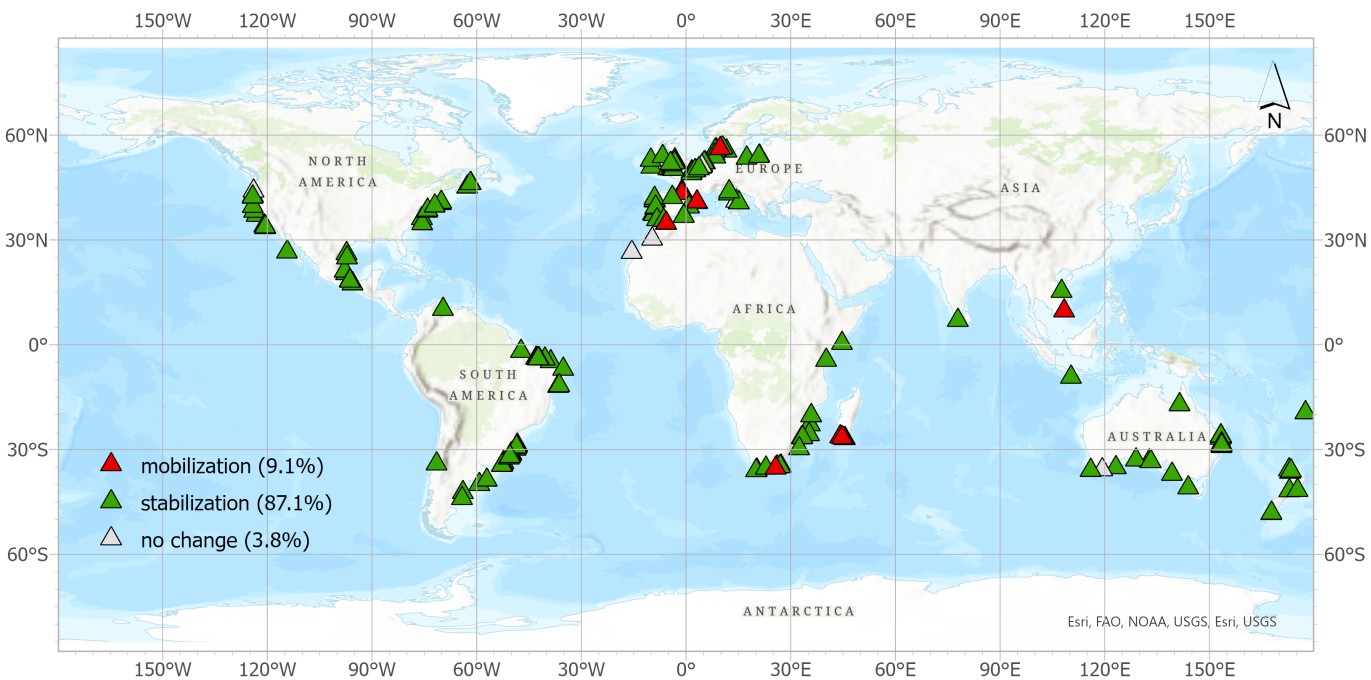

**Figure 4.** Dune dynamics based on Mann–Kendall analysis performed on the trend component of the NDVI series.

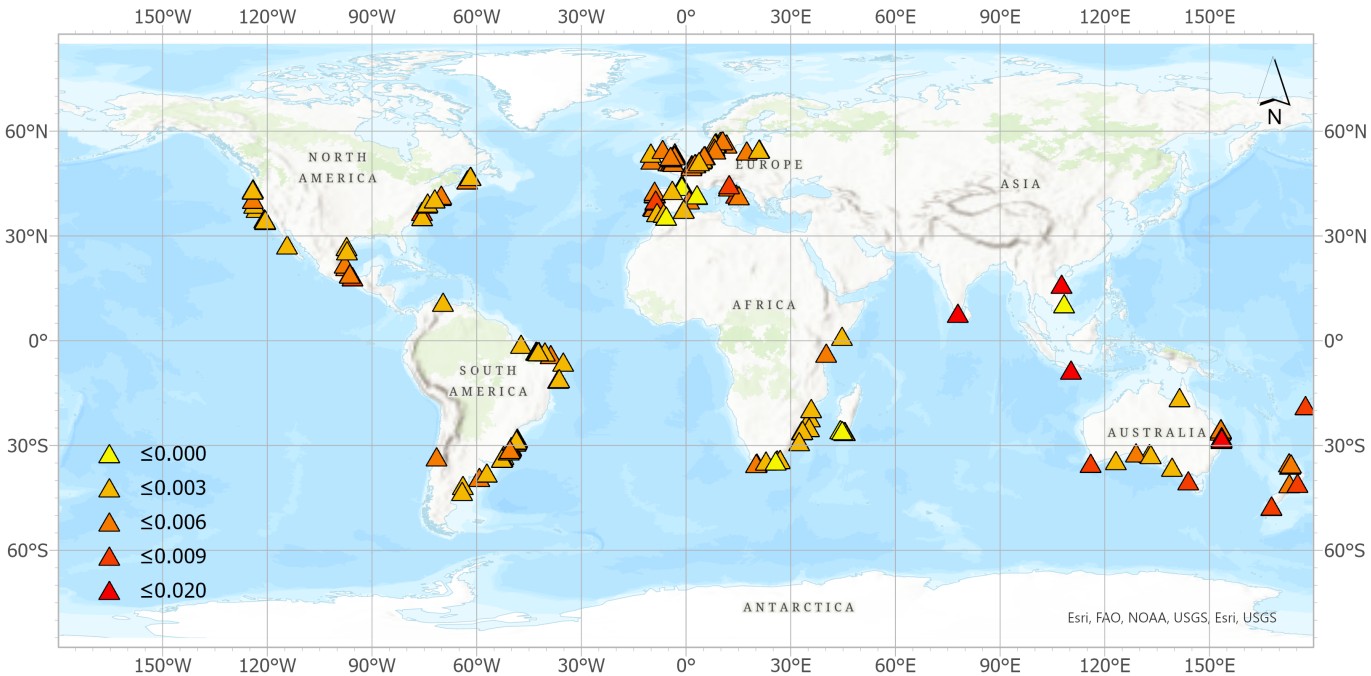

**Figure 5.** Sen's slope (per year) based on the trend component of the NDVI series. Only sites with a statistically significant nonzero slope are shown. Positive slopes are split into classes that correspond to slowly increasing ($0 < Q_{NDVI} \leq 0.003$ per year); moderately increasing ($0.003 < Q_{NDVI} \leq 0.006$ per year); rapidly increasing ($0.006 < Q_{NDVI} \leq 0.009$ per year); and (4) very strongly increasing ($Q_{NDVI} > 0.009$ per year) greenness.

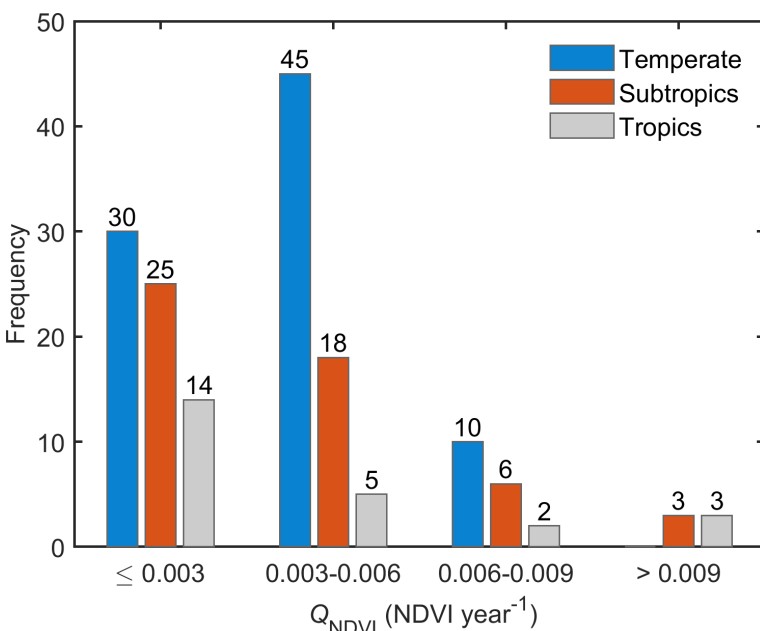

**Figure 6.** The frequency of various classes of Sen's slope, $Q_{NDVI}$, for different latitudes (temperate, between 35° and 66.5°; subtropics, between 23.5° and 35°; and tropics, between 0° and 23.5° north/south latitudes).

To further examine the intersite variability in coastal dune greening, we explored whether there was a dependence of $Q_{NDVI}$ on the temporal mean greenness, $\overline{NDVI}$, here defined as the temporal mean of the trend component. The $\overline{NDVI}$ in our data set of greening and mobilizing dune sites, visualized in Figure 7, varied between 0.049 for the Guerrero Negro barrier islands in Mexico (arid conditions) and 0.605 for Brownslade Burrows in Wales (cool temperate climate), with a data set mean of 0.290. From all dune sites with $\overline{NDVI}$ within the largest 25% of the data set, approximately a third originates from Europe. A visual comparison of Figures 5 and 7 shows that greening took place for the entire range of $\overline{NDVI}$; in other words, there was no $\overline{NDVI}$ threshold that separated greening from mobilizing dunes. Next, the frequency distribution of $Q_{NDVI}$ was constructed conditional on $\overline{NDVI}$ to provide further information on their dependence. To this end, the greening sites were classified as sites with predominantly bare soil or sparse vegetation ($\overline{NDVI} \leq 0.3$) and largely vegetated surfaces ($\overline{NDVI} > 0.3$), while for $Q_{NDVI}$, the four rate categories defined in the previous paragraph in conjunction with Figure 5 were used. It can be seen in Table 1 that 57% (51 out of 89) of the sampled dune sites with no or limited vegetation had some of the lowest trend slopes ($0 < Q_{NDVI} \leq 0.003$ per year), while the rate of change for 58% (42 out of 73) of the well-vegetated dunes was moderately large ($0.003 < Q_{NDVI} \leq 0.006$ per year). This suggested that larger $Q_{NDVI}$'s were associated with larger $\overline{NDVI}$'s. A Chi-square test of independence confirmed this association. The $\chi^2$ value for the data in Table 1 was 17.86, which is well above the critical $\chi^2 = 11.34$ for independence with $\alpha = 0.01$ and three degrees of freedom. This reaffirmed the inferences drawn from Figure 6, as most sites with a high $\overline{NDVI}$ were from the temperate zone (between 35° and 66.5° north/south latitudes, Figure 7).

**Table 1.** Contingency table for the dependence of Sen's slope $Q_{NDVI}$ on the mean greenness $\overline{NDVI}$.

| $Q_{NDVI}$ / $\overline{NDVI}$ | 0–0.003 | 0.003–0.006 | 0.006–0.009 | >0.009 | Total |
|---|---|---|---|---|---|
| ≤0.3 | 51 | 27 | 8 | 3 | 89 |
| >0.3 | 18 | 42 | 10 | 3 | 73 |
| Total | 69 | 69 | 18 | 6 | 162 |

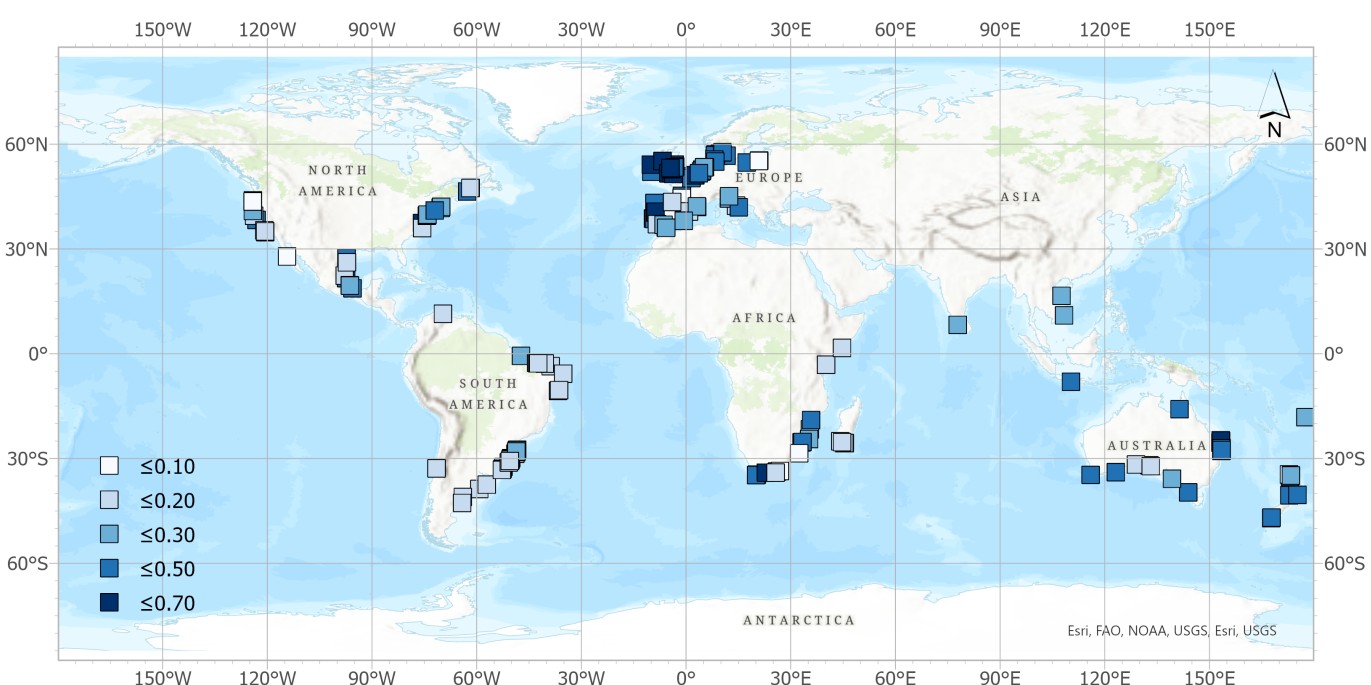

**Figure 7.** Temporal (1984–2021) mean of the trend component in NDVI.

The intersite variability in greening was also compared with the strength of the seasonal component in NDVI, denoted here as ΔNDVI and calculated as the range between the 10th and 90th percentile of the zero-mean seasonal component Se. A joint visual inspection of the maps of ΔNDVI (Figure 8) and greening (Figure 4) revealed that greening also took place irrespective of the strength of the seasonal NDVI. The largest ΔNDVI's were found in dune fields in cool to warm moist conditions, most notably in Northwestern Europe and on the East coast of the US (Figure 8). In Northwestern Europe, ΔNDVI varied between 0.103 at Anholt Island in Denmark and 0.298 at Brownslade Burrows in Wales. The lowest ΔNDVI value in our data set was reached for the dry conditions of Guerrero Negro, Mexico (=0.009), and the maximum value was 0.345 for the Greenwich dunes in Canada with cool temperate moist conditions. As expected, ΔNDVI depended on $\overline{\text{NDVI}}$. The majority of greening sites with $\overline{\text{NDVI}} \leq 0.3$ had ΔNDVI values less than 0.1 (76 out of 89 sites; 85%), while ΔNDVI at sites with $\overline{\text{NDVI}} > 0.3$ mostly had ΔNDVI values in the range of 0.1–0.2 (34 out of 73 sites; 47%) and 0.2–0.3 (22 out of 73 sites; 30%).

In the results presented so far, we examined greening and its rate for the entire length of the time series, even though the example NDVI time series in Figure 3a already revealed substantial nonstationarity in the trend (Figure 3b). Therefore, a further trend analysis of all greening and mobilizing dune sites was performed by comparing $Q_{\text{NDVI}}$ values over two 20-year periods: the first counted from the beginning of the NDVI time series and the second counted from the end. This approach resulted in a small overlap, because the study period was less than 40 years. Some of the records were shorter due to lacking observations over the first years of measurements. Moreover, these records were split into two parts of equal duration; the shortest time series (six sites only; among them, Ovari Beach, India) had approximately 20 years of data, thus 10-year-long periods were compared. The results are provided in Table 2, in which $Q_{\text{NDVI}}(+)$ and $Q_{\text{NDVI}}(-)$ represent greening and mobilizing dunes, respectively. As can be seen, the greening persisted at the vast majority of sites that greened during the first part of the study period (133 out of 140). What is more, at 86 of these 133 sites, the magnitude of $Q_{\text{NDVI}}$ increased, implying an acceleration in the greening. Figure 9 illustrates that sites with accelerating greening can be found in all climate zones, but that they are especially sparsely vegetated sites ($\overline{\text{NDVI}} < 0.3$) located in the tropics and subtropics (Figure 9b). Table 2 further shows that of the 39 sites that initially mobilized, only 8 remained so, with 6 of these 8 sites having a less negative slope in the

second part of the study period. At the remaining 31 sites, $Q_{NDVI}$ changed sign, implying a shift into greening.

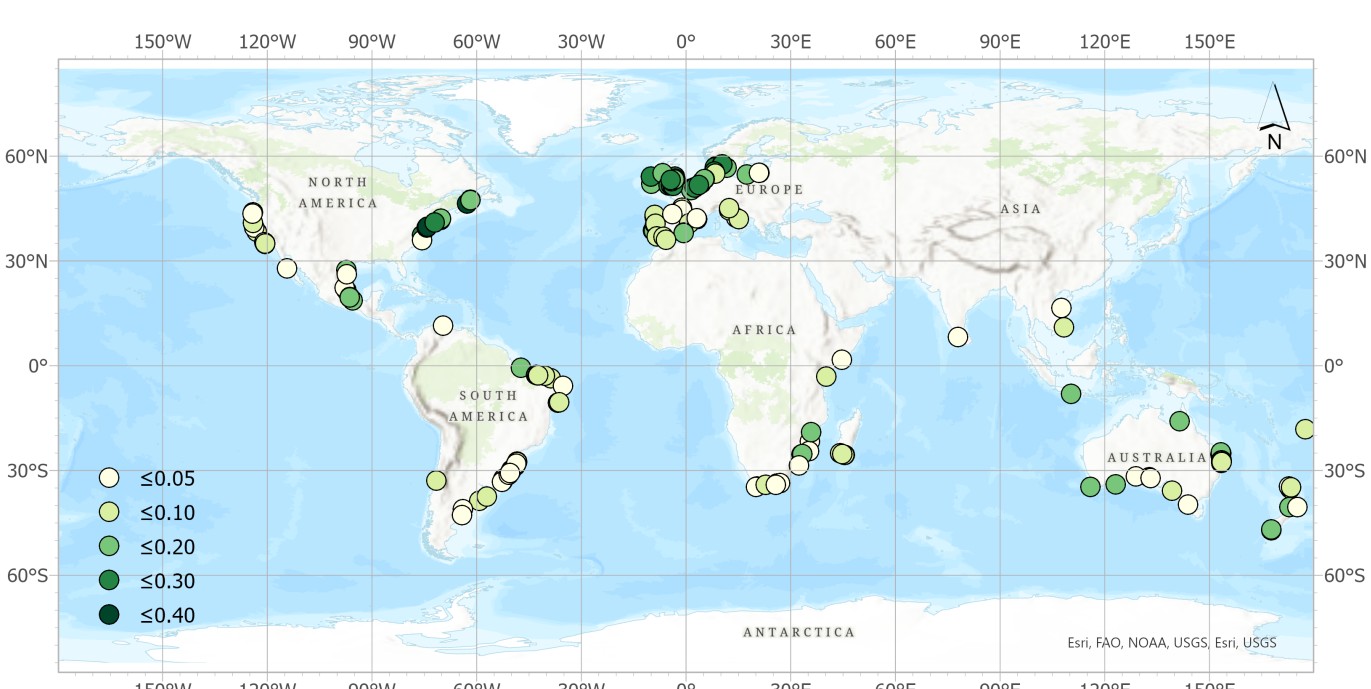

**Figure 8.** Strength of the seasonal component ΔNDVI for the study period 1984–2021.

**Table 2.** Dune cases for each combination of trend directions ($Q_{NDVI}$) over the first and second part (typically 20 years each) of the study period. $Q_{NDVI}(+)$ and $Q_{NDVI}(-)$ represent greening and mobilizing dunes, respectively. Only sites with statistically significant $Q_{NDVI}$ during the entire study period were considered.

| First Part | Second Part | Total | Accelerated | Decelerated |
|---|---|---|---|---|
| $Q_{NDVI}(+)$ | $Q_{NDVI}(+)$ | 133 | 86 | 47 |
| $Q_{NDVI}(+)$ | $Q_{NDVI}(-)$ | 7 | 2 | 5 |
| $Q_{NDVI}(-)$ | $Q_{NDVI}(-)$ | 8 | 2 | 6 |
| $Q_{NDVI}(-)$ | $Q_{NDVI}(+)$ | 31 | 23 | 8 |

*3.2. Climate Drivers of Dune Greening*

Trends in the time series of daily mean temperature (at a 2 m height) $T$ (°C), daily mean wind speed (at a 10 m height) $W$ (m/s) and accumulated daily precipitation levels $P$ (mm/day) and their relationship with dune greening are explored in this subsection. The analysis was based on the assumption that natural dune mobility is controlled mainly by changes in the local climate [75]. Therefore, if that is the case, it should be expected that greening dune fields are correlated with warmer, wetter and less windy conditions. The cases with increased aeolian activity, on the other hand, should couple with changes in the climate favouring sand transport. The first step in the analysis was the statistical testing for significant trends ($p \leq 0.01$) in the climatic time series for the 162 greening and 17 mobilizing dune sites. Table 3 shows that the mean daily temperature increased with time for both greening and mobilizing sites, with only few exceptions. In contrast, a similar number of dune cases with statistically significant positive and negative trends for the total daily precipitation and the mean daily wind speed were found (Table 3).

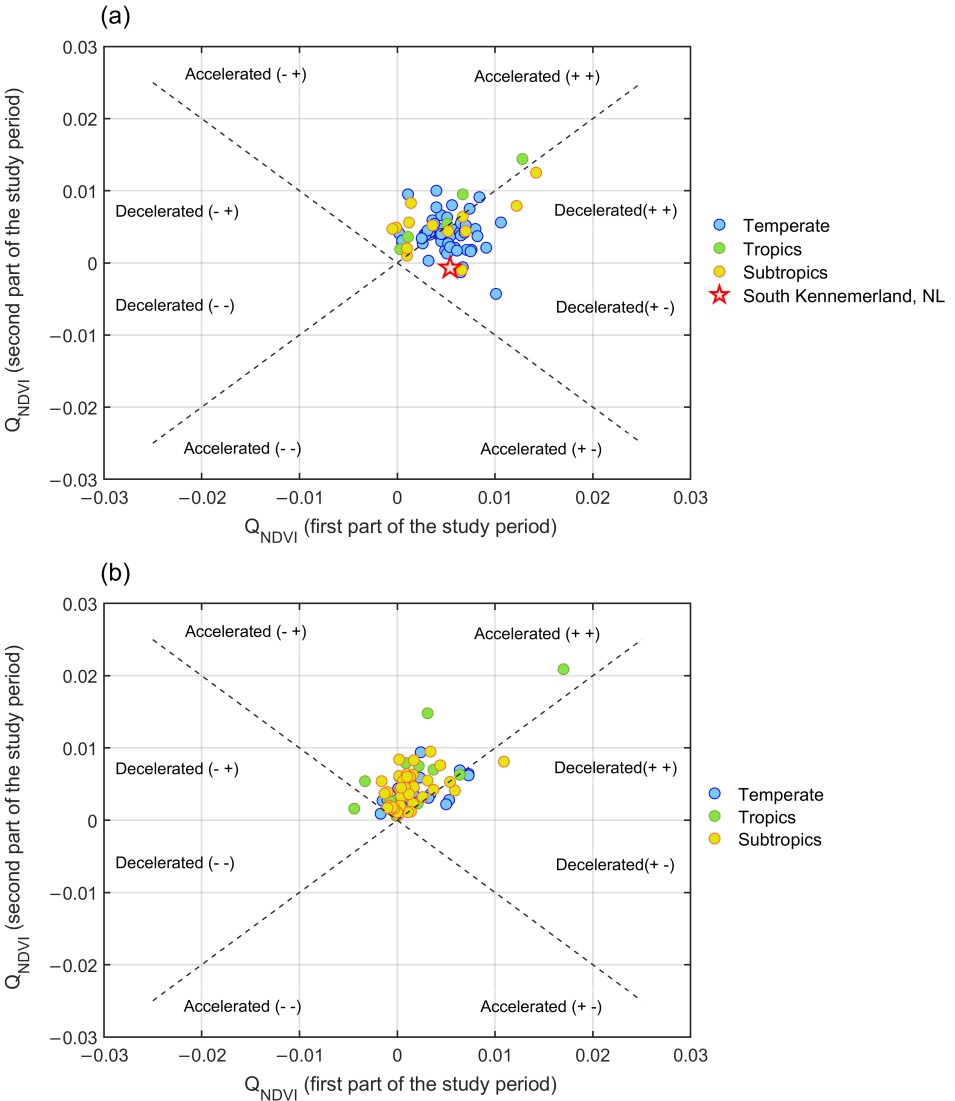

**Figure 9.** Sen's slope $Q_{NDVI}$ during the second versus the first part of the study period for sites with (**a**) $\overline{NDVI} > 0.3$ and (**b**) $\overline{NDVI} \leq 0.3$. In each panel, the data are split into the three climatic zones also used in Figure 6. Only sites with a statistically significant positive $Q_{NDVI}$ during the entire study period are shown. The $\overline{NDVI}$ separation is based on the entire study period. Panel (**a**) also shows the values of the South Kennemerland study site of Figure 3, for which we noted a change from a positive into a negative NDVI trend over time; see also Section 4.

**Table 3.** Mann–Kendall test results ($\alpha = 0.01$) for the daily climate variables

| Variable | # Significant $Q > 0$ | # Significant $Q < 0$ | # Not Significant $Q = 0$ |
|---|---|---|---|
| Greening dune sites (162) | | | |
| Average daily temperature $T$ | 158 | 1 | 3 |
| Accumulated daily precipitation $P$ | 67 | 63 | 32 |
| Average daily wind speed $W$ | 59 | 63 | 40 |
| Mobilizing dune sites (17) | | | |
| Average daily temperature $T$ | 16 | 0 | 1 |
| Accumulated daily precipitation $P$ | 7 | 10 | 0 |
| Average daily wind speed $W$ | 8 | 6 | 3 |

Figure 10 shows scatterplots of $Q_{\mathrm{NDVI}}$ (greening sites only) versus the rate of change in $T$, $P$ and $W$ ($Q_T$, $Q_P$ and $Q_W$, respectively), together with the results of a linear regression analysis. For all three climate variables, the square of the correlation coefficient, $R^2$, was very low (<0.02), and the $p$ values were well above $\alpha = 0.01$, implying that the rate of greening did not depend statistically significantly on the temporal change in temperature, precipitation or wind speed. A multiple linear regression model was also not statistically significant ($p = 0.142$; $R^2 = 0.034$). Similarly, a multiregression analysis for dune sites that mobilized also showed insignificant results.

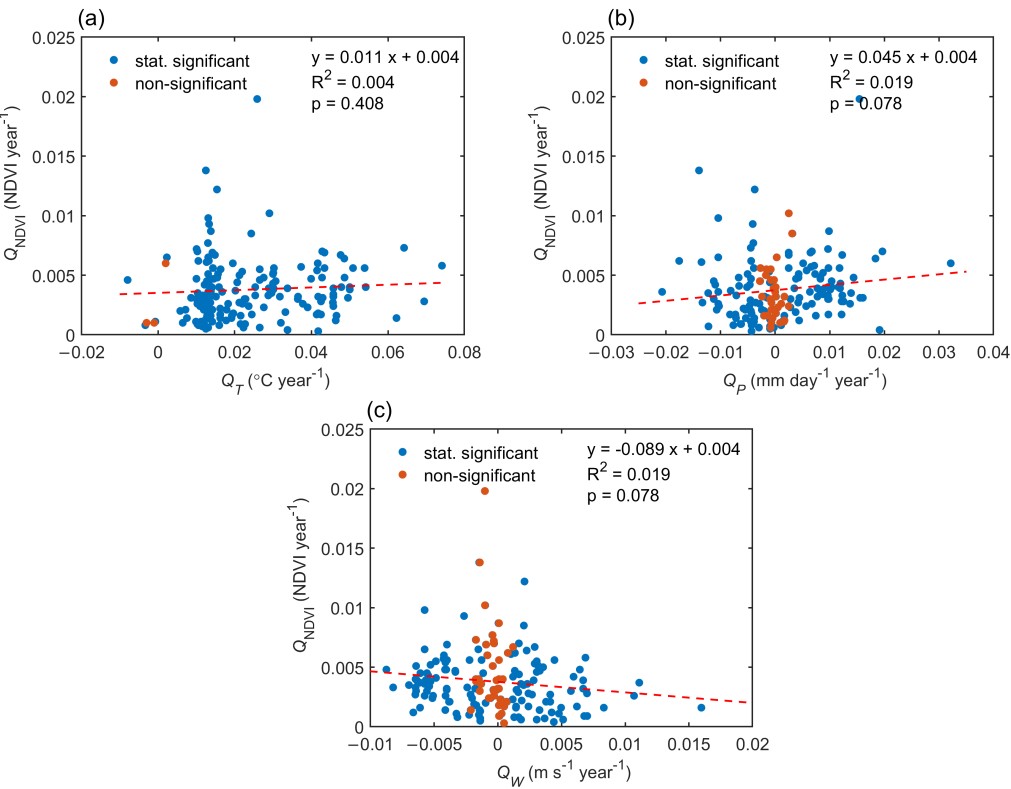

**Figure 10.** Sen's slope $Q_{\mathrm{NDVI}}$ versus the Sen's slope in (**a**) temperature $Q_T$, (**b**) precipitation $Q_P$ and (**c**) wind speed $Q_W$ for the greening dune sites. Blue and brownish dots indicate whether Sen's slope in the climate variables is statistically significant, see also Table 3. The red line in each panel is the best-fit linear line, with details given in the upper right. None of these lines is statistically significant at $\alpha = 0.01$.

Because the (multiple) regression analyses did not return any statistically significant dependence between the rate of greening and temporal changes in the considered climatic variables, we next explored whether greening always happened in the same classes of favourable combinations of local trends in the meteorological parameters ($Q_T$, $Q_P$ and $Q_W$). In particular, the climatic conditions stimulating vegetation expansion are described in the most general case by $Q_T > 0$, $Q_P > 0$ and $Q_W < 0$, which, for simplicity, are denoted as $T\uparrow$, $P\uparrow$ and $W\downarrow$, respectively. It should be noted that, for the period of study (1984–2021), the daily temperature increased at nearly all greening sites (Table 3). Other possible combinations of local climatic conditions included: (1) $T\uparrow$, $P\uparrow$, $W\uparrow$ (warmer, wetter, windier); (2) $T\uparrow$, $P\downarrow$, $W\downarrow$ (warmer, drier, less windy); and (3) $T\uparrow$, $P\downarrow$, $W\uparrow$ (warmer, drier, windier). The last combination promotes aeolian activity, while the first two may eventually also do so if the sand drift potential is high enough [76].

Table 4 shows the distribution of the greening dune sites with $T\uparrow$ across the combinations of trends in the local meteorological parameters. Approximately one-third of these greening dune sites (50 out of 158) were associated with the combination of conditions that favoured greening ($T\uparrow$, $P\uparrow$ and $W\downarrow$). These sites were mainly from cool temperate climates,

among which Northern Europe and the East Coast of the USA. Table 4 also reveals that a substantial number of greening dune sites (43 out of 158) were associated with climate conditions for which an increase in aeolian transport was expected ($T\uparrow$, $P\downarrow$ and $W\uparrow$). The same analysis was also performed for the mobilizing sites with $T\uparrow$. Table 4 shows that six of these sites (37.5%) indeed had $T\uparrow$, $P\downarrow$ and $W\uparrow$; these included the chevrons on Madagascar and dune fields in NE Brazil. Other mobilizing sites, including Dune du Pilat in France, Terschelling in the Netherlands and Long Beach Island in the USA, were, however, associated with climate conditions that were expected to favour greening ($T\uparrow$, $P\uparrow$ and $W\downarrow$; 4 out of 16, Table 4). On the whole, while (regional) climate variability may have led to dune greening or mobilization in some locations, this relationship is clearly not global. This is examined further in the Discussion section.

**Table 4.** Greening and mobilizing dune sites grouped according to change in temperature, precipitation and wind speed.

| Conditions | # Cases | Interpretation |
|---|:---:|---|
| Greening dune sites with $T\uparrow$ (158) | | |
| $T\uparrow, P\uparrow, W\downarrow$ | 50 | Greening |
| $T\uparrow, P\uparrow, W\uparrow$ | 30 | Mobilizing/greening |
| $T\uparrow, P\downarrow, W\downarrow$ | 35 | Mobilizing/greening |
| $T\uparrow, P\downarrow, W\uparrow$ | 43 | Mobilizing |
| Mobilizing dune sites with $T\uparrow$ (16) | | |
| $T\uparrow, P\uparrow, W\downarrow$ | 4 | Greening |
| $T\uparrow, P\uparrow, W\uparrow$ | 3 | Mobilizing/greening |
| $T\uparrow, P\downarrow, W\downarrow$ | 3 | Mobilizing/greening |
| $T\uparrow, P\downarrow, W\uparrow$ | 6 | Mobilizing |

## 4. Discussion

The global tendency for greening of coastal dunes (Figure 4) was in line with the literature review in [29] for a broad range of geographically spread dune sites and with the findings in [22], who also based their analysis on NDVI data from Landsat imagery. The present study used a much larger sample of dune fields than [22] and additionally quantified through various indices derived from the NDVI time series that the increase in greening was to a great extent site-specific (Figures 7 and 8). Ref. [29] also noted site-to-site differences in their literature review but did not quantify the associated greening rates. We illustrated that well-vegetated dunes, distinguished by $\overline{\text{NDVI}} > 0.3$, were mainly associated with moderately steep trends (0.003–0.006 per year), while sites with a permanently low vegetation cover $\overline{\text{NDVI}} \leq 0.3$ had some of the lowest temporal changes ($\leq 0.003$ per year). An important result of the present work was that many greening sites experienced a larger increase in NDVI in the last two decades of the studied time interval compared to the first two decades (Table 2 and Figure 9). Coastal dune greening thus appeared to accelerate. What is more, most of the sites that mobilized during the first two decades started to green in the subsequent two decades (Table 2).

Contrary to expectations based on the climate-based mobility concept for natural dunes, e.g., [77,78], we found no dependence between the rate of greening and the temporal change in temperature, precipitation and wind speed. Neither the individual correlations nor the multiple regression analysis returned statistically significant dependencies ($p$ values well above 0.01; Figure 10). Moreover, the results of the grouping of greening and mobilizing sites according to changes in temperature, precipitation and wind speed were inconclusive (Table 4). For the greening cases, for example, a roughly equal number of sites (50 versus 43) were associated with conditions that would result in greening ($T\uparrow$, $P\uparrow$, $W\downarrow$) and in mobilization ($T\uparrow$, $P\downarrow$, $W\uparrow$). Thus, there was not a clear global link between coastal dune greening and climate variability.

Our work confirmed earlier findings, e.g., [11,29] that changes in dune mobility cannot be attributed to a single driver. Based on the existing literature, we now briefly list possible other causes for intersite variability in dune greening together with a number of limitations that were inherent to our work and could have affected the results.

* Sand supply: A few of the investigated dune cases here demonstrated a substantial mobility due to excessive sand supply combined with strong local winds. The sand had either a local origin (e.g., Rubjerg Knude dune, Denmark; dunes on Isle de Madeleine, Canada), or was brought to the dune site through alongshore drift (the Maranhão dunefields, NE Brazil). At the Rubjerg Knude dune, the local climate may support either mobilization or greening, depending on the relative importance of wind and precipitation ($T \uparrow$, $P \uparrow$, $W \uparrow$); however, [79] argued that strong local winds were responsible for active transport of the abundant sand eroded from the cliff and the inland migration of the dune. Unlimited sediment supply, combined with drier conditions and very strong winds, may be responsible for the here identified mobilization of the Bolonia and Valdevaqueros dune systems (SW Spain) and of Duna de Cresmina (Portugal). In contrast, an increased presence of nebkha dunes in arid conditions indicates a reduced sediment supply [39,80], here reflected in the slowly greening trend at Guerrero Negro.

* Management: Anthropogenic activities may have impacted dune vegetation cover and the dune dynamics at many sites significantly [29], with certain human activities intended for stabilization (e.g., planting, afforestation; [9,81,82]), and others leading to sand mobilization (e.g., introduction of grazers [43,83]; dune rejuvenation programs [20,82]). In general, previous European management practices included widespread stabilization to protect the coast from flooding, with marram grass planting and afforestation to stop migrating sand [9,81,82]. Indeed, the present study corroborated that some of the largest key indices describing dense dune vegetation with a marked seasonality corresponded to dune sites from N and NW Europe. There, many sites also experienced warming temperatures and declining wind speeds. On the other hand, greening in NW European coastal dune sites has also been ascribed to increased atmospheric nitrogen deposition [84,85] and the drastically reduced population of rabbits [86,87]. The latter was also the dominant cause for the greening of the dunes on the Younghusband Peninsula in Australia [12]. Dune management at several sites selected in the present work has recently adopted an approach that aims at stimulating dune mobility. Particular cases of mobile dunes analysed here correspond to management attempts to reactivate the foredunes by removing the vegetation cover. An example is the Dutch barrier island of Terschelling, where the foredune was reactivated in the mid 1990s, and a significant sand burial of the more landward dune area has taken place since [18]. This was detected here in the negative trend of the analysed NDVI time series, although the local climate trends were expected to promote greening. Moreover, the change from increasing to decreasing NDVI over time in Figure 3b (see also Figure 9a) was due to a dynamic restoration project, here carried out in 2012–2013 [20]. Again, the mobilization persisted (at least until 2021) despite climate conditions favouring vegetation growth.

* Climate state: It is possible that the current climate at several sites has already crossed the threshold of being sufficiently warm, wet or still for coastal dunes to green. This may imply that at such sites further changes in the climatic variables are irrelevant to greening. In other words, sites may not respond simultaneously or in the same way to climate change. The latter may also be caused by the inherent hysteresis of dune mobility to vegetation cover [76]. The same magnitude of change in wind speed may thus affect dune sites with different vegetation cover (in this paper expressed as $\overline{NDVI}$) in different ways.

Finally, we note that we focused on quantifying the effects of regional climate on dune mobility, while assessing the importance of other factors in the preceding paragraphs only qualitatively. Ref. [11] provided a semiquantitative procedure to split the effects of climate

change and other factors quantitatively. Therefore, our investigation could be extended in future work to also quantify the magnitude of the nonclimatic perturbations. This would make it possible to evaluate what the direction of change in the dune system would have been if the other disturbances had been removed. This extension would, however, require the estimation of the climate-induced vegetation cover, for which, to the best of our knowledge, no quantitative methods yet exist.

## 5. Conclusions

Based on the analysis of NDVI time series from 1984 up to and including 2021 computed from Landsat imagery for 186 coastal dune sites, we conclude that greening dominates coastal dunes worldwide, regardless of the temporal mean greenness $\overline{\text{NDVI}}$ or of the strength of the seasonal NDVI component. This confirms earlier studies that were based on fewer sites. The rate of greening $Q_{\text{NDVI}}$ depends on $\overline{\text{NDVI}}$ such that larger $Q_{\text{NDVI}}$'s are associated with larger $\overline{\text{NDVI}}$'s. Accordingly, dune sites in the temperate climate zone ($35°$ to $66.5°$ north/south latitudes) tend to green faster than the generally less green sites in the subtropics and tropics. Moreover, greening has accelerated over time at many locations worldwide, especially at sparsely vegetated sites ($\overline{\text{NDVI}} < 0.3$) in the tropics and subtropics. Second, we conclude that there is no statistically significant dependence of the rate of greening on the local temporal change in temperature, precipitation and wind speed. While this may be due, at least in part, to inherent limitations of our approach, the existing literature on dune mobility illustrates that other natural and/or human-induced factors, including sand supply and management policies, can dominate vegetation growth. At a number of locations, the anthropogenic impact on vegetation cover outweighs the role of the local climate, at least temporarily.

**Supplementary Materials:** The following supporting information can be downloaded at: https://www.mdpi.com/article/10.3390/rs15061491/s1. Table S1: Site names, together with the latitudes and longitudes.

**Author Contributions:** Conceptualization, G.R.; methodology, formal analysis and visualization, P.G.P.; writing—original draft preparation, P.G.P. and G.R.; writing—review and editing, P.G.P., G.R. and S.M.d.J.; supervision, G.R. and S.M.d.J.; project administration, G.R. and S.M.d.J. All authors have read and agreed to the published version of the manuscript.

**Funding:** This research received no external funding.

**Data Availability Statement:** All NDVI and climate time series were downloaded from the Climate Engine research app (https://app.climateengine.com/climateEngine [65]); accessed most recently on 24 February 2023. The app is licensed under a Creative Commons CC-BY license. The decomposition code was downloaded from https://github.com/Ghaderpour/LSWAVE-SignalProcessing; accessed most recently on 24 February 2023.

**Acknowledgments:** We thank Didier Haagmans, whose BSc research under the supervision of S.M.d.J. and G.R. initiated the MSc work of the first author. We are grateful to the four reviewers whose comments helped us to better highlight the innovative parts of the work and bring clarity to the methods employed.

**Conflicts of Interest:** The authors declare no conflict of interest.

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
