# Peer review of "A Global Remote-Sensing Assessment of the Intersite Variability in the Greening of Coastal Dunes"

_remotesensing, doi:10.3390/rs15061491_

Round 1

Reviewer 1 Report

Comments to the manuscript entitled “A Global Remote-Sensing Assessment of the Greening of Coastal Dunes”.

After carefully reading and considering all points in the manuscript, I regret to conclude that I cannot see what are the new findings that it brings compared to the two previous works published in the topic and also referred by the authors. The authors even adopt the terminology proposed by one of the manuscripts and choose a title that is too close to the one mentioned. The authors claim that they were able to prove that global ongoing climate change is not the main driver of the observed increase of vegetation at a global scale, a conclusion already reached by Gao et al. (2020), who mention the climate as a secondary factor after a diverse list of human activities that promote the observed change. Yet, the approach chosen by the authors of the present manuscript to show the lack of relation between climate variables and observed changes might not be the more adequate as they start comparing both type of observations, forgetting that changes may not be simultaneous at all. Then, they decide to compare with the direction of the trend, without considering the magnitude of those trends or the relevance of the observed trends to the evaluated systems, meaning, do the changes in wind are enough to promote a drop or enhance sediment transport?

In addition, the authors have included in their analysis sites that compromise the credibility of their assessment and interpretations. One example is the Dune du Pilat in France. This system is well known because of its very large size that results from the vertical stacking of different units over time. It is also very well known to tourists and visitors as the larger dune in EU where trampling and other activities are allowed and promoted, which in turn prevent the vegetation to grow and ensure the mobility of the dune. So, there is not doubt that dune mobility there is allowed because of the human pressure and not because of the large amount of sediment. The authors suggest that this dune is moving because of that large amount of sand, and despite the drop of wind, according to third authors cited in the manuscript. However, the paper they refer only shows the results of a model that is able to simulate and reproduce the migration of transverse and NON VEGETATED dunes. So, why do the authors choose to select such a site? How can the authors prove their point regarding the impact of climate change with such cases? It looks to me that using such examples to support that climate has not impact is not the more appropriate way to proceed as some of the chosen examples are not allowed to evolve.

Therefore, and when seeing to their total numbers, I see that 85 cases out of 174 coincide with an increase in temperature and a decrease in wind, which could favour the sealing of dunes, explaining about 70% of the cases. So, I think climate seems a good candidate. However, in order to bring something new into the discussion, the actual implication of these changes at a local scale should be assessed. I believe that if the authors wish to bring something new they should analysis closer cases to actually understand if the changes in climate variables may or not have an actual impact on the observed dunes, otherwise, I regret to tell that this work is a sort of a repetition of the previous ones focusing on this problem. Besides, the authors mention that the next step will be to link human actions to the observed changes, so, I suggest the authors to focus on this in order to actually contribute with something new.

I also would like to explain to the authors that the use of the “greening”, applied to coastal dunes firs by Jackson et al. 2019, derives from a recent work focusing on the greening of continents at a global scale. In this work, the authors suggest that the main cause behind this change was the increase in CO2, yet, the authors do not discuss this problem at all within their work, which is on the other hand strange if they aim to prove the lack of connection between global change and the vegetation cover.

Finally, I would like to add that the conclusion by the authors that vegetation expansion is faster in higher latitudes seems likely related to the fact that both, climate change and human activities, are more intense in this latitudes, so, again pointing towards the climate and human activities, the problem is how can we split these two factors? Is that possible?

Below I add few more points, per section, that I believe the authors should further consider revising if they decide to improve their work.

Abstract:

The authors mention “…, potentially negatively affecting the dunes’ resilience to climate change…”. This affirmation, even shown as a potential, is not supported by any work to the reviewer knowledge, so, I recommend to avoid using such affirmations, which also include the second part of the sentence, have the authors examples of the fact that vegetation increase decreases diversity? This is a rather strange affirmation when reading the title of the work and having in mind that they are presenting areas that were dominated by bare sand and are becoming vegetated…so, how can they affirm that this colonization can reduce diversity?

The last sentence seems to show what is actually new, however, these three points are discussed very briefly and not at a site scale as they should to actually prove what the authors suggest.

Introduction:

I regret to tell that the authors here totally forget about the different morphologies that dunes may present around the world, and to the importance of those to the erosion and evolution of the dune profile, which may also depend on the vegetation, but maybe mostly to the geomorphology.

The statement “Furthermore, the lower mobility can reduce biodiversity and hence cause the dunes to be less able to withstand stressors imposed by climate change.” needs references or maybe, this should be further investigated before it can become a statement.

The authors state “On the whole, it remains unclear whether coastal dune greening is a global trend resulting from global climate change or is a site-specific response to local to regional anthropogenic and/or climatic conditions.”, this is very odd affirmation again, it seems clear that the greening is a global trend, I have not seen proves of the contrary. Yet I do agree with the authors that the reason is not clear as there are some authors suggesting climate as the main driver and others human impact. Yet, what I cannot understand is the suggestion that greening can be driven by regional climate conditions as those cannot be disconnected from the global scale, so, which is the open question that the authors pretend to answer?

The authors intend to compare the observations with hindcast time series of climatic variables. This is a very good idea, but it must be done carefully as systems may not react simultaneously to the climatic variables. Also, the magnitude of the changes might have a different impact depending on the actual dynamics of each system, how is this going to be considered?

Material and methods:

What do the authors mean by “relatively natural dunefields”?

Results:

Looking at figure 6; The fact that you find that temperate regions vegetated more than the others is not related to climate change?

Would you still need to use the seasonal component if you only assess the vegetation change of a particular season?

Table 3 presents the comparison between trends as if the magnitude of the change was not important. I believe this must be particularly relevant to the wind trends and the impact to the potential at a local scale.

Discussion:

Lines 322-324: the affirmation is correct to Jackson et al. 2019, but not to Gao et al. 2020. In the previous works it was already clear that the rates of greening were different from site to site, as the interactions of all the drivers determining the % of vegetation are not linear and may change depending on local conditions. Also, the authors here do not explore the nutrients and the human activities. Pilat dune is mobile because people are invited to run on top of it!! How could vegtt grow?

Line 330 “Coastal dune greening thus appears to accelerate”. This seems a very interesting point of your outputs and maybe it is also related to the acceleration of changes in factors over time?

Lines 331-333, this cannot be concluded using dunes that are artificially maintained as mobile.

Line 338-339: this seems also not easy to assess because assuming that the impact of the rate of change in temperature and wind have the same impact on the dune may not be straight forward conclusion.

Reviewer 2 Report

Dear Editor

I thought it would be easier to come up with a list of points to consider instead of introducing notes in the original text.

Notes for the authors:

1- What were the criteria for choosing the 186 study sites? Were the studied dune fields that experienced significant erosion and/or accretion between 1984-2021 excluded from the study?

2- Using a 30 x 30 m pixel what is the criterion used to define the dune field/beach boundaries?

3- line 385 areas with “dynamic restoration project” such as Dutch barrier island of Terschelling were included in the study? The intervened areas do not describe the intended natural pattern of evolution (1984-2021).

4- Figure 2. Why the authors do not characterize South Kennemerland (Netherlands) since it is used as "truth data" in Figure 3 and 9. And explore/focus this information further in 3.1. NDVI variability and Figure 9 (a)

5- The results shown in Figures 4 and 5 should be made available in the supplementary data

6- lines 173- 175, the authors note that “Finally, we note that we will use the phrase ‘mobilization’ rather than ‘browning’ to express that a negative NDVI trend implies a significant increase in the number of sand pixels (i.e., in aeolian dynamics) and not in a deterioration of the vegetation greenness.” But in line 195 use “greenness” but in Figure 5 and 6 the authors don´t use it. And in line 215 the authors start using mobilizing versus greening . It would be important to have a standardization of the terms used.

7- Figure 5- include in the legend the intervals of the QNDVI  tal como aparece no texto ( (1) slowly increasing (0 < QNDVI 0.003 per year); (2) moderately increasing (0.003 < QNDVI 0.006 per year); (3) rapidly increasing (0.006 < QNDVI 0.009 per year), and (4) very strongly increasing (QNDVI > 0.009 per year) greenness)

8- In Figure 5 and 6 QNDVI values are presented but the word greenness is never used. Its use would facilitate the reading of the results. In figure 6 it would be important to have the designation of each class (slowly increasing; ...)

9- Figure 7- are missing the classes of “trend component in NDVI classes”

10- Figure 8. Strength of the seasonal component ∆NDVI, classes are missing

11- Table 2- it is important to include the meaning of  QNDVI(+) and QNDVI(-) represent greening and mobilizing dunes

12- line 252; why 20-year periods? Why did the authors not do the analysis considering sites with identical climate? This information seems to be presented in Figure 9 but is unclear in the text;

13-Line 266-268 it is important to review the text because it is not very explicit

14-  line 279 which “17 mobilizing dune sites” and the climate conditions?

15- The study should be site specific this weakness is particularly visible in section 3.2. Climate drivers of dune greening. Is it not possible to do the same analysis by choosing one example for each climate type?

16- Line 327 the authors use the term NDVI as a synonym for lowest temporal changes. The authors can clarify this point from the beginning.

17- Line 328 and 329 “the larger increase in NDVI with time in the last two decades of the studied time interval compared to the first two decades, irrespective of the initial direction of change (Fig. 9).” Please clarify this sentence by subdividing the time series

18- Line 339 the sentence “There is thus not a clear global link between coastal dune greening and climate variability. "is perhaps too strong since it is known that more important than precipitation (P) is moisture/humidity a determining factor in the development of coastal dune vegetation and sediment transport and even more site-specific than precipitation.  Do the authors have any information on relative humidity/ moisture at the site South Kennemerland (Netherlands) that could serve as validation data?

19-When analyzing dune fields in very different climates it becomes difficult to frame the evolution of dune vegetation cover. In temperate climates the variability of NDVI is greater because it varies with the seasons. It would be important to remove the effect of short-lived herbaceous plants that cover the dunes in spring after the rains. In fact this vegetation cover is temporary and does not describe an evolution of the dune field.

20-In order to eliminate seasonal variations, one could use the catalog of reflectance curves of the most common dune plants at each of the studied sites to correct the NDVI value

Reviewer 3 Report

This paper reports important and interesting methodology and results on monitoring of the greening of worldwide coastal dunes. The manuscript is well organized and the novelty of the study can be clearly recognized, and thus recommended for the publication in this journal.

One concern is the resolution of the image data. 

The resolution of Landsat images is ~30m. 

Therefore, I guess that the dunes with width smaller than 100m could not be included in the data sets of this study.

As far as I know, however, a majority of coastal dunes are small as their width may not clearly larger than 100 m (correct if I am wrong).

Therefore, I recommend that the limitations of the availability of the data sets due to the resolution of the satellites be added in the manuscript, along with the expectation caused by that limitation. 

In addition, it is necessary that the detailed procedure to determine the ROI when the size of the vegetation was comparable to the satellite resolution, as such an example is provided in Figure 2b.             

Minor:

Figure 2: the spatial scales and the resolutions of Landsat need to be inserted/marked in the figures so that the size of ROI can be recognized, compared to the satellite resolution.

In addition, the ROI in Figure 2a is not clearly detectible in the picture, so need to be revised for the ROI to be easily recognized.  

Reviewer 4 Report

     In recent years, with the global temperature rising, the increase of vegetation greening has attracted many researcher's attention. This study focuses on the greening of coastal sand dunes, and collects as much information as possible about sand dunes around the world to clarify the greening of sand dunes, and tries to clarify the changes of sand dune greening and its relationship with its activity. At the same time, it also discusses the control factors of sand dune greening, such as temperature, sand supply and human activities.

 Although the discussion of these control factors is not deep and comprehensive enough, the study provides some new progress and new perspectives on the study of coastal dunes. If this research can conduct a comprehensive and in-depth study of the specific impact factors of each studied beach especially the rules of the impact of enhanced human activities on the greening of sand dunes, we can provide more guidance for the management of sand dunes.

Round 2

Reviewer 1 Report

Comments to the reviewed version of the manuscript “A Global Remote-Sensing Assessment of the Greening of Coastal Dunes” now entitled “A Global Remote-Sensing Assessment of the Intersite Variability in the Greening of Coastal Dunes”.

The authors have made a very good work improving the manuscript, which is much clearer now regarding for instance the objectives and findings. In this regard, I must congratulate them and accept the manuscript with the substantial changes they have addressed. Yet, I am still concern about the adequacy of some of the selected sites as the answer from the authors was not very convincing. This is the case again of Pilat dune. The authors mention that the dune is still active because of the elevated availability of sand and refer to papers focusing on the Holocene dunes (so, not very adequate) and a mention to the work by Provoost that refers to the work of Tastet, and then, to the Holocene origin of the dunes, so, I understand this feature is recycling from the former one. Then the same author (Provoost) also stresses the fact that the dune may also remain active because “dune building can be influenced by human interference in the physical or biological environment”, which could help to explain why is so confined laterally. Anyway, I do believe that this example is very tricky and would recommend avoid using it. Instead, the authors could use a dune, also active donwdrift Pilat as it presents a more “natural” plant shape.

For the rest, I find the manuscript improved and better focused so I will recommend its publication after considering the Pilat issue by the authors.

Author Response

We are pleased to see that the reviewer finds our revised paper much improved and suitable for publication after we have addressed the remaining concern about Dune du Pilat. 

We agree with the reviewer that the reasons for the mobilizing character of Dune du Pilat are trickier than we initially thought. Therefore, in line with the reviewer's suggestion, we have removed Dune du Pilat from the Discussion section of our paper as an example of a dune system where sand supply rather than climate change dominates mobility (lines 362-366). This subsection still contains several undisputed examples, so the deletion of Dune du Pilat does not harm the discussion here.

Reviewer 3 Report

No additional comments. 

Author Response

We thank the reviewer for reading our revised paper. We are glad to see that our rebuttal has answered all comments on the original paper version and that there are no new comments.